# Rapid increase in the risk of heat-related mortality

Samuel Lüthi ®[1,2] ✉, Christopher Fairless[1], Erich M. Fischer ®[3],
Noah Scovronick ®[4], Ben Armstrong ®[5],
Micheline De Sousa Zanotti Stagliorio Coelho ®[6], Yue Leon Guo ®[7,8,9],
Yuming Guo ®[10], Yasushi Honda[11], Veronika Huber ®[12,13], Jan Kyselý[14,15],
Eric Lavigne[16,17], Dominic Royé ®[18], Niilo Ryti ®[19], Susana Silva ®[20],
Aleš Urban[14,15], Antonio Gasparrini ®[5,21,22], David N. Bresch ®[1,2] &
Ana M. Vicedo-Cabrera ®[23,24] ✉

Heat-related mortality has been identified as one of the key climate extremes posing a risk to human health. Current research focuses largely on how heat mortality increases with mean global temperature rise, but it is unclear how much climate change will increase the frequency and severity of extreme summer seasons with high impact on human health. In this probabilistic analysis, we combined empirical heat-mortality relationships for 748 locations from 47 countries with climate model large ensemble data to identify probable past and future highly impactful summer seasons. Across most locations, heat mortality counts of a 1-in-100 year season in the climate of 2000 would be expected once every ten to twenty years in the climate of 2020. These return periods are projected to further shorten under warming levels of 1.5 °C and 2 °C, where heat-mortality extremes of the past climate will eventually become commonplace if no adaptation occurs. Our findings highlight the urgent need for strong mitigation and adaptation to reduce impacts on human lives.

Extreme heat is associated with substantial impacts on human health[1–4]. In the past, extreme heatwaves in under-prepared communities have been responsible for several thousands of deaths within just a few weeks[5–7]. With anthropogenic climate change already accounting for roughly a third of heat-related deaths[8], the risk of deadly heat is projected to further increase as the climate continues to warm rapidly[9–11]. While several studies have projected future heat-related mortality, they were based on a selected set of deterministic scenarios, and estimated future heat mortality levels as the mean of a given climate period or scenario[12–14]. These approaches do not capture the full suite of possible climate futures and may under-represent the potential risk for heat-related mortality, which can be driven by rare but extreme years. At the same time, probabilistic projections of extreme heat often lack the additional step of quantifying the corresponding human impacts, including the health impacts[15,16]. To allow for a more comprehensive risk assessment of highly impactful events for human

health, we apply a probabilistic approach to the quantification of future heat-related mortality risk. Such probabilistic risk assessments are widespread in the risk assessment of natural hazards, such as flooding[17] or tropical cyclones[18], as knowledge of the magnitude and probability of potential impacts are key to prepare for and adapt to climatic extremes[19]. Our analysis is performed for 748 locations from 47 countries for which the Multi-Country Multi-City (MCC) Collaborative Research Network collected observed daily mortality and temperature data during recent decades (Table 2). Using this data, we quantify empirical exposure-response functions[20] which we combine with output from five single-model initial-condition large ensembles (SMILEs)[21] by using the natural catastrophe risk platform CLIMADA[22]. The SMILE climate model output is generated by running a single climate model multiple times with perturbed initial conditions but following the same radiative forcing scenario. This creates diverging weather and climate patterns for each model run, resulting in an

ensemble spread that displays the internal climatic variability. The variability within the ensemble allows us to explore physically plausible extreme years and is thus well suited to estimate tail-risks.

This study quantifies not only the magnitude of potential future heat-related excess mortality, but also the frequency, which can be valuable to decision makers, as the ability to withstand climatic extremes is often based on past experience[23,24]. Our use of the city (or small region) as the unit of analysis is also administratively relevant for adaptation planning[25,26]. We focus on heat, since this is the emerging risk that health departments need to prepare for, but note that cold-related mortality is higher throughout many locations in our data set.

## Results

### Heat-mortality associations and exceedance frequency curves

We first modelled the relationship between mean daily temperature and mortality in each of the 748 locations (Table 2). The relationships are expressed as relative risk and interpreted as the change in mortality risk at specific temperature values against an optimum temperature (the so-called temperature of minimum mortality, MMT)[8,13,20]. As described in more detail in the Methods, these functions reflect the complex relationship between temperature and all-cause mortality by accounting for the delayed and nonlinear impact that heat has on human health. The exposure-response functions are then used to calculate the fraction of deaths attributable to heat based on the estimated risk corresponding to the mean temperature value on each day. Heat-related mortality corresponds to the average fraction of daily deaths attributed to heat during days with mean temperature above the MMT. As Fig. 1a–c shows, temperature-mortality associations tend to be U-shaped, but still differ (sometimes substantially) from city to city, including in the MMT, which is generally higher in hotter cities. To derive the probabilistic projections of heat-related mortality, we hold these estimated relationships constant for all time periods, which enables a straightforward comparison of the potential impact of different levels of warming on mortality, but does not account for demographic changes (especially population ageing) or adaptation. We hence compute heat mortality by multiplying the day-of-year average mortality counts from the empirical data in each location by the relative risk associated with the (projected) temperature for that day and summarize it to annual levels (Methods). This approach estimates heat-related deaths while also preserving the annual cycle of mortality. The same method was used in foregoing studies to derive heat-mortality estimates for different scenarios and study periods[8,12].

To characterize probabilistic mortality impacts, we plot impact exceedance frequency curves, which relate the magnitude of impacts (here annual heat-mortality fractions) to its frequency of occurrence. Specifically, we express the frequency through a "return period", defined as the inverse of the cumulative occurrence probability. A 1-in-100 year heat-mortality level—i.e. a 100 year return period—thus refers to the mortality that is exceeded by 1% of modeled years within a given climate period for each of the five SMILEs. We do this annually for every model for the four 20-year climate periods that represent mean warming levels in the year 2000 (0.7 °C of warming above a 1850–1900 reference period), the year 2020 (1.2 °C warming), as well as for 1.5 °C and 2.0 °C of warming. This approach hence yields up to 1680 (depending on the number of ensemble members per SMILE) equally probable years for each climate period and model.

The impact exceedance frequency curves at each of the four different warming levels are displayed for three selected locations, representing different continents and climate zones, in Fig. 1d–f. Heat-related mortality in 2003 in Paris, which included the record breaking European summer, amounted to 5.9% (95% CI: 4.7–7.3%) of total annual mortality (an estimated 2718 (2142–3371) deaths in the city), a level expected to occur only about once every 100 years in the climate of

2000 (Fig. 1e). However, we find that in the climate of 2020, the same mortality impacts would be expected every 18 years (model IQR: 16.6–20.4), which is in line with previous studies[27]. At 2.0 °C, this level of mortality would be the norm and expected to occur every few years if no adaptation to extreme heat occurs.

In 2014, São Paulo experienced heat-related mortality of 1.7% (0.7–2.8%) of total mortality, or 1296 deaths (556–2095, in line with ref. 28), a burden that would be expected every 134 (67–217) years in the climate of 2000; the return period decreases to 18 years (17.0–19.6) in the climate of 2020, 11 years (8.0–13.1) at 1.5 °C and 5 years (2.7–5.5) at 2.0 °C (Fig. 1d). Return periods for Bangkok similarly shorten with increasing levels of warming.

To summarize, the impact exceedance frequency curves consistently project a rapid reduction in return periods of heat-related mortality; the mortality fractions experienced during past extreme years should be expected much more frequently (shift along the x-axis). The curves also show that heat impacts increase, both for frequent (1-in-10) and extreme (1-in-100) years (i.e., shift along the y-axis). Figure 1g–i reports the estimated mortality fraction from heat that would occur in a 1-in-100 year for each SMILE separately. This fraction would more than double in all three example cities if global mean temperatures rose from 0.7 °C to 2.0 °C above pre-industrial levels, albeit with uncertainty due to different warming rates of the climate models, internal climate variability (inner whisker of the grey ticks) and the epidemiological uncertainty within the relative risk associations (outer whisker), as displayed by the shaded area (95% CI) in Fig. 1a–c. The internal climate variability is assessed by bootstrapping the ensemble members for each model-specific exceedance frequency curve. Overall, these results show that ongoing, rapid shifts in mean and extreme temperatures limit the utility of past observations for assessing the risk of even present-day or near-future heat risks.

### Changes in return periods across the globe

In Fig. 2, we display the new return period for all 748 locations of the annual heat-mortality level of a 1-in-100 year season in the climate of 2000. Despite regional differences, the results show a strong shortening of return periods throughout the globe. Changes are especially pronounced on the US Atlantic and Gulf coast as well as the Latin American Pacific coast, the Mediterranean region, the Middle-East and South-East Asia. The changes in return periods over Europe show a North-South gradient with stronger shifts in Southern Europe, despite more pronounced warming over Northern Europe[29]. This impact gradient is also present in related studies[13] and highlights the importance of incorporating location-specific exposure-response relationships into this risk analysis. Uncertainties in vulnerability are generally higher in locations where the annual temperature range is less than 10 °C, including and especially in (sub-)tropical locations. This induces larger uncertainties when extrapolating the relative risk curve to higher temperatures (see e.g. Bangkok, Fig. 1c).

Overall, our results show that the potential for increased risks from heat-related mortality is not only a concern for the future, but something that has already manifested over the past two decades. In addition, Fig. 2 highlights how limiting long-term warming to 1.5 °C would entail a substantially lower risk than what is projected for a 2 °C world. Even so, for most locations, the extreme years of the past will become commonplace in the near future, requiring substantial adaptation to avoid large-scale harm.

### Uncharted territories ahead

In addition to the changes in the frequency of extreme years, the increases in the magnitude of mortality of low probability (1-in-100 year) seasons is also of high relevance to societies and decision makers. Heat-mortality during extreme seasons can amount to more than 10% of total deaths in several locations even under moderate climate

scenarios, according to the multi-model mean (Fig. 3), which represents a doubling or even tripling of mortality impacts during extreme seasons. The risk is especially pronounced throughout Europe, South-East Asia and the Latin American Pacific coast. Furthermore, locations with a historically low burdens of heat mortality are projected to suffer potentially high death counts during an extreme season, again highlighting the value of a probabilistic risk perspective as these types of extreme seasons are of special relevance to societies. Still, for locations in Central America, inland US, North Western Europe and South Africa, the risk is less pronounced, even under climate change. However we note that the latter (South Africa) may be due in part to data artifacts[30] and warn against interpreting those results as definitively implying low-risk.

## Worsening the odds of heat-mortality

We want to highlight two main findings from our study. First, we found that what used to be extreme (1-in-100 year) heat-mortality seasons are becoming frequent, and need to be expected every 2–5 years in most locations. Second, with non-linear effects, both in new climatic extremes, as well as in the vulnerability of communities to heat, unprecedented impacts on populations health need to be expected when assuming no adaptation. We found that mortality levels in Paris that were expected to be exceeded once in 100 years in 2000 are exceeded 5 times in 100 years in the 2020 climate, 10 times in a 1.5 °C world and 27 times in a 2 °C hotter world (Fig. 4). This increase in probability of mortality levels is even larger for seasons that were highly unlikely in 2000 (1-in-500 year seasons) as they need to be

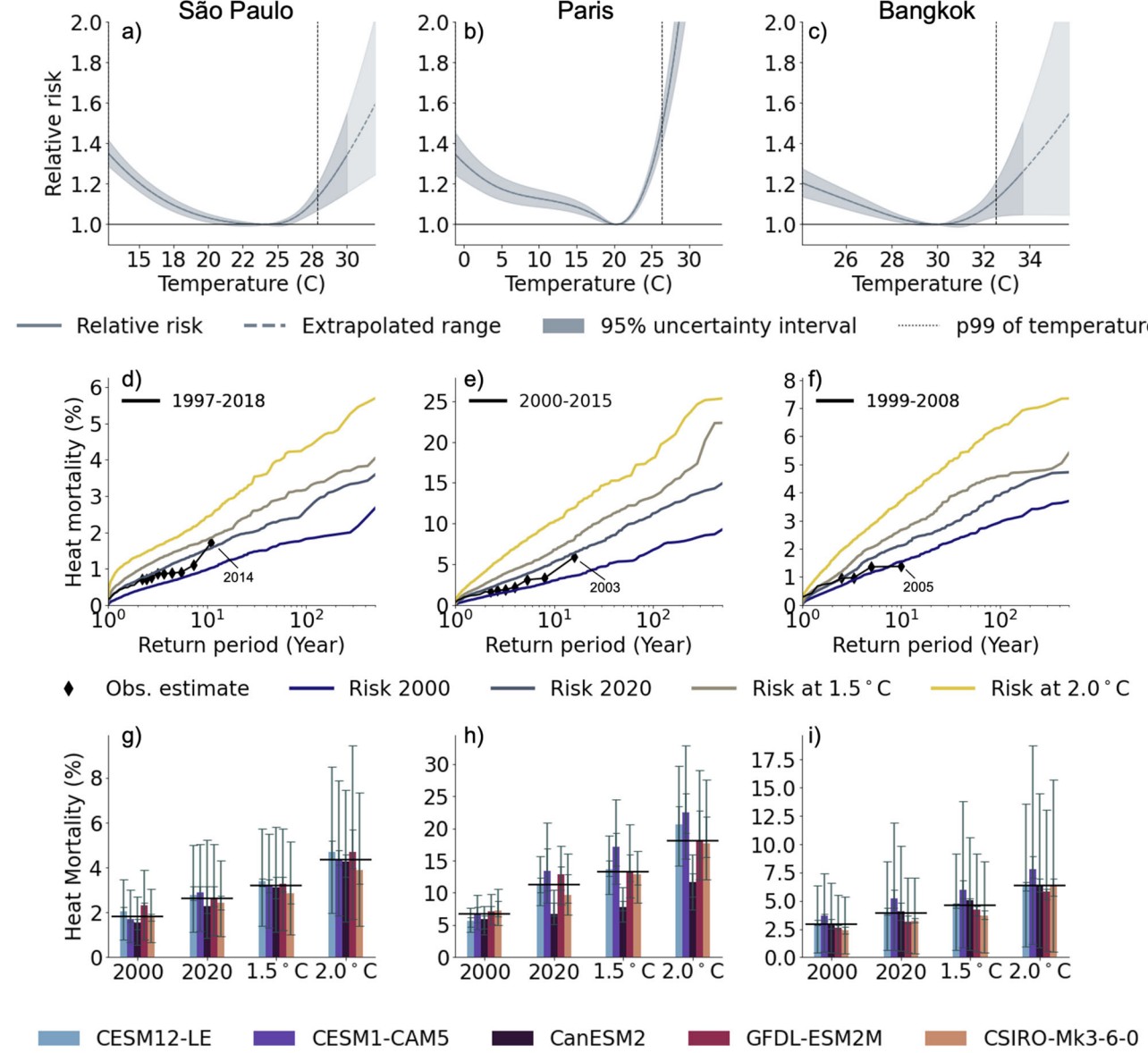

**Fig. 1 | Risk of heat mortality for São Paulo, Paris and Bangkok.** Risk of heat mortality for São Paulo (Brasil, **a**, **d**, **g**), Paris (France, **b**, **e**, **h**) and Bangkok (Thailand, **c**, **f**, **i**). **a–c** Relative risk of mortality relative to the location-specific minimum mortality temperatures reported as best linear unbiased predictions (BLUPs) with 95% confidence interval (shaded area). Vertical dotted lines show the log-linear extrapolation used for projections when future temperatures exceed current temperatures. Dashed vertical lines show present-day 99th percentile temperatures. **d–f** Impact exceedance frequency curves of annual heat mortality fractions for the observed years (black line, markers denote individual years), as well as the climate of 2000 (warming level of 0.7 °C), the climate of 2020 (warming level of 1.2 °C), 1.5 °C warming and 2 °C warming. The modelled impact exceedance frequency curves are reported as the median value over the five single-model initial-condition large ensembles (SMILEs). **g–i** Modelled magnitude of the annual heat mortality fraction from a 1-in-100 year season for different warming levels. Results are displayed for each SMILE (bars) including uncertainty estimates depicted with the 95% empirical confidence intervals accounting for the internal climate variability (inner whisker) and imprecision of the exposure-response associations (outer whisker). The black horizontal line denotes the median estimate for each global warming level.

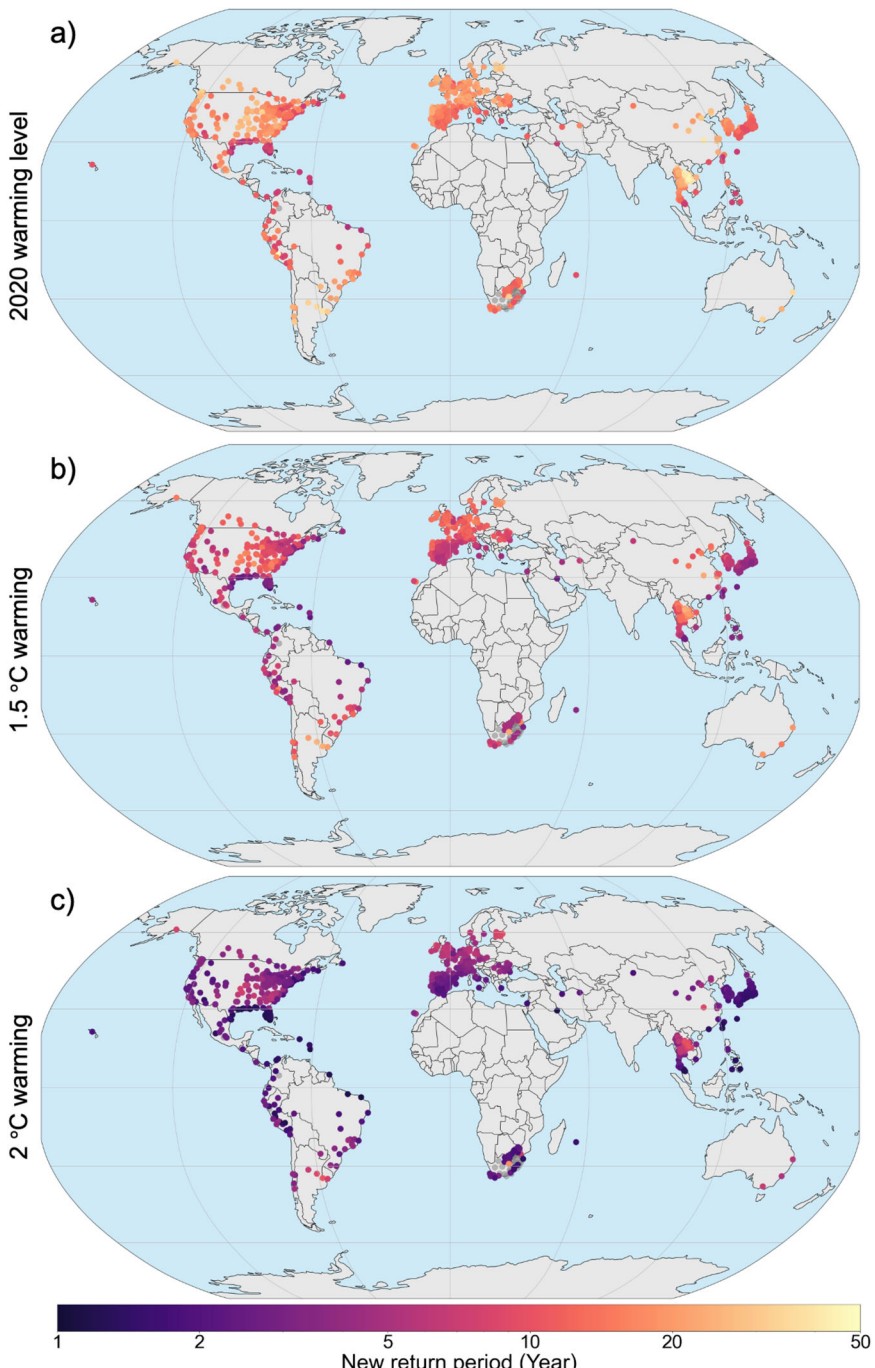

**Fig. 2 | Changes in return periods of a 1-in-100-year season in the 748 locations.** Changes in return periods for the climate of 2020 (warming level of 1.2 °C, **a**), at 1.5 °C warming (**b**) and at 2.0 °C warming (**c**) compared to the risk in the climate of 2000 (0.7 °C warming). The figure displays the new return period of the location-specific 1-in-100-year heat-mortality level of 2000. The colour-scale is logarithmic. The grey dots denote locations with inconclusive results due to their spread in uncertainty.

expected 14 times in 100 years in a 2 °C hotter world, thus increasing their likelihood by a factor of 69. Hence, even under warming levels in line with the Paris agreement (1.5–2 °C), non-extreme seasons are becoming increasingly rare for most locations while uncharted territories are first becoming the new extremes and then eventually regular.

## Discussion

In this work we combined state-of-the-art techniques from climate change epidemiology with the latest approaches in climate science to quantify extreme seasons. Our results align well with related work

although a direct comparison of numbers is hampered by different selection of reference scenarios, time scales or geographical scope[27,31]. As an example, the change in return period of the mortality counts during the 2003 heatwave in Paris was estimated to decrease from 1-in-300 years, in a world with no anthropogenic climate change to a 1-in-70 years in the actual climate of 2003[27], which aligns well with our findings (Fig. 1). Looking solely at climatic variables, the rapid reduction of return periods of such heatwaves were reported early on[32], estimating the 2003 temperature anomaly over Southern Europe to occur every other year (1-in-2 years return period) by 2040. Similarly, Christidis et al.[33]. found a tenfold decrease of the return period of extremely hot

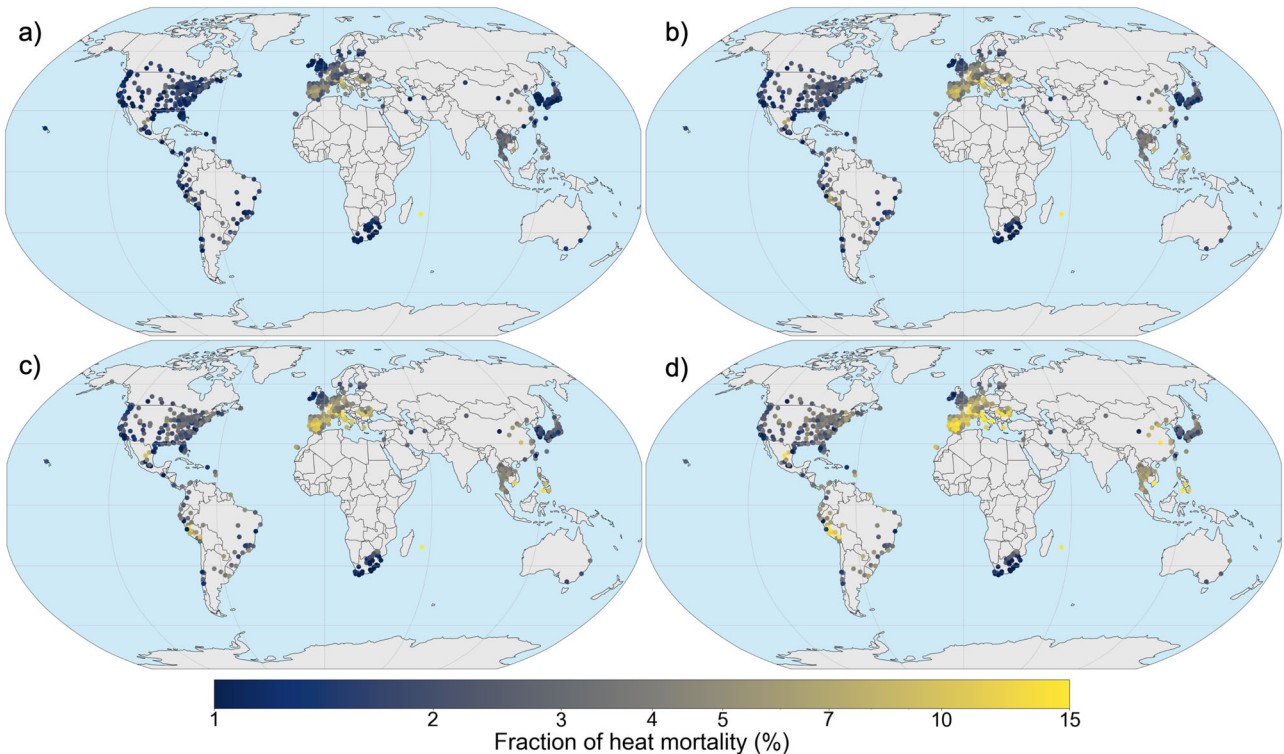

**Fig. 3 | Heat-mortality fraction of a 1-in-100-year season in the 748 locations.** Heat-mortality fraction of a 1-in-100 year season. Rates are displayed for the climate of 2000 (**a**), 2020 (**b**), 1.5 °C (**c**) and 2.0 °C warming (**d**). For each location, shares are calculated as the heat-mortality counts during a 1-in-100 year season divided by the mean annual mortality. The colour-scale is logarithmic.

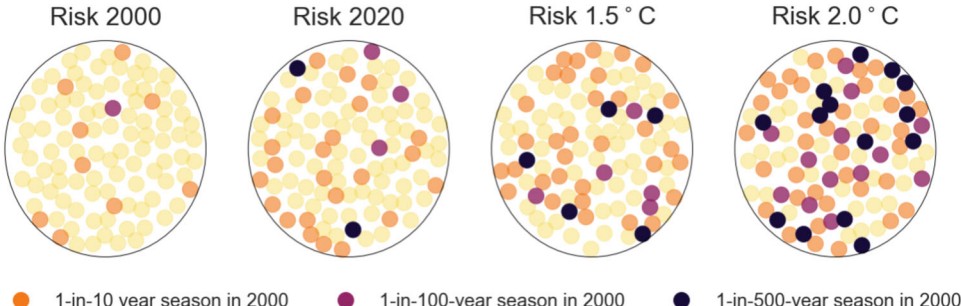

**Fig. 4 | Schematic display of internal variability of heat related mortality for Paris (France).** Each circle contains 100 points representing the climatic variability of the given warming level. The points denote the return period based magnitude of mortality of the climate of the year 2000 for a 1-in-10 year season (orange), a 1-in-100 year season (violet), a 1-in-500 year season (black) and more frequent seasons (yellow).

European summers between the 1990s and the early 2000s (from 1-in-50 years to 1-in-5 years), again aligning well with our findings. Similar results were obtained using SMILE data[34], where the return period of a monthly temperature extreme of a 1-in-100 year event in a 1.5 °C world is reduced to a roughly 1-in-10 year event in a 2 °C world, whilst the new 1-in-100 year events of the 2 °C world represent unchartered territories. Our analysis identifies several regions with a pronounced shortening of return periods of extreme seasons. For tropical regions this is largely due to the small seasonality and year-to-year variability of temperature which therefore leads to large shifts in return periods of extremes in a hotter climate[35]. In Southern Europe, Japan and along the US Atlantic and Gulf coast, the projected reduction in return periods is furthermore driven by demographic influences, such as the aging of societies and the heightened vulnerability of populations to heat.

These results highlight a need to incorporate possible extreme scenarios and storylines of unprecedented heatwaves into the planning of public health policies as the experience from mortality impacts of past summer season is likely to underestimate the actual risk of heat mortality in the rapidly changing climate. Currently, most (European) heat-health warning systems focus on issuing warnings to relevant authorities and vulnerable people during or slightly before the onset of a heatwave[36]. However, in contrast to other climatic extremes, such as floods, only few authorities systematically plan for rare but extreme seasons[36].

We acknowledge some limitations of this study. First, despite having access to what is, to our knowledge, the most comprehensive data set for climate change epidemiology, our geographical scope is somewhat limited and overrepresents Western regions compared to other parts of the world. This is potentially problematic, as severe heat is expected to increase strongly in many tropical regions (especially also in Western and Eastern Africa and India[11]) with highly limited (financial) capacity for adaptation. However, given the highly location-specific risk-response curves, we refrained from extrapolating to these regions. In addition, impacts are analyzed for each location as a whole,

**Table 1 | Properties of single model initial condition large ensemble (SMILEs) used in this study**

| Climate Model | Number of members | Model resolution | Forcing | Model time period | | | |
| --- | --- | --- | --- | --- | --- | --- | --- |
| | | | | 2000 (0.7 °C) | 2020 (1.2 °C) | 1.5 °C | 2.0 °C |
| CESM1.2 | 84 | 1.9° × 2.5° | Hist/rcp8.5 | 1989–2008 | 2006–2025 | 2014–2033 | 2027–2046 |
| CESM1-CAM5 | 40 | 1.3° × 0.9° | Hist/rcp8.5 | 1990–2009 | 2007–2026 | 2015–2034 | 2028–2047 |
| CanESM2 | 50 | 2.8° × 2.8° | Hist/rcp8.5 | 1982–2001 | 1995–2014 | 2003–2022 | 2016–2035 |
| GFDL-ESM2M | 30 | 2.0° × 2.5° | Hist/rcp8.5 | 1988–2007 | 2010–2029 | 2024–2043 | 2042–2061 |
| CSIRO-Mk3.6.0 | 30 | 1.9° × 1.9° | Hist/rcp8.5 | 1996–2015 | 2014–2033 | 2022–2041 | 2035–2054 |

Table properties partly adapted from Deser et al.[21].

which leaves us agnostic to intra-community differences in vulnerability due to age[5,37], race/ethnicity[38,39], gender[40] or poverty[37,41]. Also important small-scale climatic differences, such as urban heat island effects are not resolved[42]. This is potentially problematic, as they can overlap with vulnerable communities[39]. Second, we use constant risk-response relationship and do not account for future adaptation. Fortunately, from a public health perspective, the evidence suggests that communities often (though not always) adapt to warming conditions[43–45]. However, for most locations the data is up-to-date and thus well-suited to display the current-day risk, although recent summer seasons are not fully covered (Table 2). The projected warming levels need to be expected to be reached in the near future—1.5 °C by 2030 and 2 °C by 2042 under the business-as-usual scenario SSP5-8.5[29]—which leaves cities little time for adaptation. Still, we therefore refrained from displaying projections for higher warming levels although the modelling set-up and data would allow to do so. Third, as mentioned above, we do not account for changes in the demographic structure of populations, such as population growth, ageing and increased urbanization. Fourth, the risk-response curves needed to be extrapolated to temperatures unobserved in the empirical data, which entails uncertainty. These points mark relevant areas for future research, especially the incorporation of changes in population as well as adaptation into future projections of heat mortality levels. Nevertheless, trends and signals remain stable (Fig. S1, Supplementary Information). Also, the stochastic uncertainty of exceedance frequency curves is well captured within each SMILE (Fig. S2, SI). The spread between different SMILEs is relatively small for past and current risk due to the applied bias-correction but increase over time, as the models underlie different warming rates (Fig. S3, SI). However, the main signals, such as changes in return periods, are remarkably stable across all SMILEs (Fig. S4, SI).

In this study, we relied on the most expansive database on weather and health (the MCC Collaborative Research Network database), covering 134 million deaths, and data output from the large-ensemble project, representing 234 climate model runs (or more than 1 TB of climate model output). We demonstrated that the probabilistic risk of heat-mortality has already increased rapidly over the past 20 years already and is projected to further increase strongly under higher levels of global warming. These findings highlight the urgent need for adaptation to heat extremes. Finally, our results clearly state that numerous lives can be saved with strong mitigation policies that keep global warming well below 2 °C, and that efforts to limit the increase to 1.5° are of greatest importance.

## Methods
### Climate model data
We used daily mean temperature data from five SMILEs: CESM1.2[16], CESM1-CAM5[46], CanESM2[47], GFDL-ESM2M[48] and CSIRO-Mk3.6.0[49], totalling 234 climate model runs (Table 1). For all models, data is available at least from 1950–2100. After 2005, all models follow the representative concentration pathway RCP8.5[50]. We used period lengths of 20 years for each climatic reference period. 20 years are a

compromise between decreasing effects of internal variability (which is better covered in standard 30 year periods) and a clear sign of change (i.e. between the climate of 2000 and 2020). In order to obtain warming-based reference periods of 0.7, 1.2, 1.5 and 2.0 °C, we calculated the mean warming of each model over all ensemble members as compared to the reference period of 1950–1969. In line with related work[51], we selected the first 20-year period in which the respective warming level is reached, including the adjustment of 0.25 °C of observed warming that occurred until 1950–1969 against a historic reference period of 1850–1900 (on the basis of the observational HadCRUT5 dataset[52]). Hence, the different SMILEs don't necessarily cover the same years for a given level of global warming, as shown in Table 1.

### Bias-correction of climate data
We bias-corrected the climate model output to align it with the observational temperature data used to calculate the heat-mortality relationships. For that, we took model data from the nearest-neighbour grid-point to each MCC location and bias-corrected it using quantile-mapping[53]. This leads to one correction function for each combination of ensemble member and location. However, to keep internal climate variability within each SMILE, the final SMILE- and location-specific correction function, is the average over all individual ensemble-member correction function. For consistency, the same location- and SMILE specific correction function was applied for each warming level.

### Mortality data
We accessed the MCC Collaborative Research Network database for daily mortality counts and observational daily mean temperature data for 748 locations (http://mccstudy.lshtm.ac.uk/). Each location represents a small-scale geographic aggregation unit (city, metropolitan area or small region). Mortality counts depict all-cause or non-external-cause mortality (ICD-9: 0-799; ICD-10: A00-R99). Temporal data coverage of locations varies between 4 and 43 years. More information and descriptive statistics of the data is displayed in Table 2.

### Assessing temperature-mortality relationships
The epidemiological analysis to assess the association between heat and mortality in each location relies on a two-stage time-series approach which is commonly used in multi-location time-series studies[8,12,54].

First, we performed quasi-Poisson regression time series analyses with distributed lag nonlinear models (DLNM) to estimate the temperature-mortality association for each location[55]. Model specification and parameterization is based on previous studies[20,56,57], as well as the choice of mean temperature as variable[20,58–60] and tested for their sensitivity[59,61–63]. Concretely, we included a natural cubic spline of time with eight degrees of freedom per year in combination with and indicator term for day of the week to account for long-term trends and seasonality. The temperature-mortality curve is then modeled with a quadratic B-spline with three internal knots placed at the 10th, 75th

**Table 2 | Summary of the mortality data for the 748 locations**

| Region | Country | Locations (n) | Data period | Total deaths ('000) |
|---|---|---|---|---|
| Australia | Australia | 3 | 1988–2009 | 1178 |
| South America | Argentina | 3 | 2005–2015 | 686 |
| | Brazil | 18 | 1997–2018 | 3895 |
| | Chile | 4 | 2008–2014 | 325 |
| | Colombia | 5 | 1998–2013 | 957 |
| | Costa Rica | 1 | 2000–2017 | 31 |
| | Ecuador | 2 | 2014–2018 | 112 |
| | France Guiana | 1 | 2000–2015 | 7 |
| | Paraguay | 1 | 2004–2019 | 48 |
| | Peru | 18 | 2008–2014 | 633 |
| | Uruguay | 1 | 2012–2016 | 154 |
| Central America | France Caribbean | 2 | 2000–2015 | 46 |
| | Guatemala | 1 | 2009–2016 | 63 |
| | Mexico | 10 | 1998–2014 | 2980 |
| | Panama | 1 | 2013–2016 | 11 |
| | Puertorico | 1 | 2009–2016 | 27 |
| North America | Canada | 26 | 1986–2015 | 3734 |
| | USA | 210 | 1973–2006 | 38,028 |
| South Africa | South Africa | 45 | 1997–2013 | 7776 |
| Southern Europe | Greece | 1 | 2001–2010 | 288 |
| | Italy | 11 | 1996–2007 | 820 |
| | Portugal | 6 | 1980–2018 | 1925 |
| | Spain | 52 | 1990–2014 | 3017 |
| Central Europe | Czech Republic | 4 | 1994–2015 | 712 |
| | France | 18 | 2000–2015 | 1754 |
| | France Reunion | 1 | 2000–2015 | 14 |
| | Germany | 12 | 1993–2015 | 3106 |
| | Moldova | 4 | 2003–2010 | 60 |
| | Romania | 8 | 1994–2016 | 951 |
| | Switzerland | 8 | 1995–2013 | 244 |
| Northern Europe | Estonia | 5 | 1997–2018 | 168 |
| | Finland | 1 | 1994–2014 | 153 |
| | Ireland | 6 | 1984–2007 | 1058 |
| | Netherland | 4 | 1995–2016 | 3050 |
| | Norway | 1 | 1969–2018 | 271 |
| | Sweden | 3 | 1990–2016 | 717 |
| | UK | 70 | 1990–2016 | 6167 |
| Middle-East Asia | Iran | 2 | 2004–2013 | 818 |
| | Israel | 1 | 1985–2020 | 351 |
| | Kuwait | 1 | 2000–2016 | 74 |
| South-East Asia | Philippines | 13 | 2011–2019 | 821 |
| | Thailand | 61 | 1999–2008 | 1802 |
| | Taiwan | 3 | 1994–2014 | 1210 |
| | Vietnam | 2 | 2010–2013 | 108 |
| East Asia | China | 14 | 2004–2006 | 1095 |
| | Japan | 47 | 1972–2015 | 39,918 |
| | South Korea | 36 | 1997–2018 | 3070 |
| Total | | 748 | | 134,433 |

and 90th percentile of the location-specific observational temperature distributions (referred to as the cross-basis function of temperature[64]). In line with previous studies, we applied a natural cubic spline with three internal knots equally distributed up to 21 days to capture the lagged response of mortality (such as short-term harvesting and long lagged associations). We selected the 21 days lag because we use all-year mortality data and not only data of the hot season. Finally, we reduced the bi-dimensional (over temperature and time) exposure-response function into a one-dimensional (temperature only) cumulative exposure-response function which expresses the location specific relative risk of mortality as a function of local daily mean temperature (Fig. 1a–c).

Second, to make full use of the hierarchical structure of the data, the location-specific coefficients from the first stage were pooled in a multivariate-metaregression model[65]. We then derived the best linear unbiased predictions (BLUPs) representing improved location-specific estimates, especially for locations with a short time series or low mortality counts. For the meta-predictors, we use country-level gross domestic product (GDP), location specific mean and interquartile range of temperature as well as a random term with clusters of cities of the same climate zone within a country[66]. Uncertainties of the temperature-mortality relationships were quantified by generating 1000 Monte Carlo simulation samples of the sets of coefficients of the BLUPs, assuming a normal distribution of said coefficients. Finally, the BLUPs were log-linearly extrapolated to cover the additional range of temperature occurring in the warming scenarios. The epidemiological analysis was performed within the R software environment using the open-source packages dnlm[67] and mixmeta[65].

## Quantifying heat-related mortality

To quantify the heat-related mortality impacts, the epidemiological analysis needs to be combined with the SMILE climate data. We computed the heat-related deaths, or deaths attributed to heat, for each location, global warming level, ensemble member and day when the mean temperature was above MMT using the method described in Gasparrini et al.[56]. and extended by Vicedo-Cabrera et al.[13]. for climate change projections. For each day, the location specific daily baseline mortality was used to extract the corresponding fraction of deaths attributed to heat using the corresponding relative risk defined by the BLUPs at the specific daily mean temperature value obtained from the SMILEs. The daily baseline mortality was computed as the average daily deaths for each day of the year within each location. The daily counts are aggregated to yearly levels and subsequently the corresponding heat-mortality fraction was computed as the percentage of heat-related deaths over the total annual mortality. The focus on annual levels is in contrast to related work which only includes the four hottest months for each location[8] but yields the advantage to capture possible climatic shifts and a prolonging of the respective hot season. We excluded 28 locations (mainly in South Africa) from the analysis (Fig. 2), since the results became inconclusive as the uncertainty range spans more than one order of magnitude of the mortality impact size.

## CLIMADA

Heat mortality impacts were calculated using the CLIMADA (CLIMate ADAptation) platform[22], available on GitHub at https://github.com/CLIMADA-project/climada_python. CLIMADA is fully open-source and -access and a well-established risk model to model impacts of natural catastrophes such as tropical cyclones[68], flood[69], windstorms[70] or wildfires[71]. The methodology for heat mortality was adopted from the R based tutorial[54] and translated to python. The new heat module is integrated into the platform to benefit from its broader functionalities such as calculation of risk metrics and adaptation options[19].

## Impact exceedance frequency curves

Return levels and return periods of heat-related mortality in each location and for each SMILE are empirically estimated in the following way: (1) Daily heat-related mortality numbers are calculated for each ensemble member using the approach described above and aggregated to annual levels (2) This yields $N = n_{member} \times 20$ annual heat-related mortality impacts for each 20-yr period. E.g. for the CanESM2

this results in $N = 1000$ (=50 × 20) possible years. (3) The empirical probability of occurrence for any given year in this period can thus be expressed as $p = 1/N$, as all of these years can be treated with equal probability of occurrence and are independent of one-another. Hence, within the CanESM2, each modelled year has a probability of occurrence of 0.1% (=1/1000) (4) The return periods are then quantified by calculating the cumulative probabilities of exceedance of impact levels: $v(x) = 1/T(x)$, where $v(x)$ is the exceedance frequency of impact $x$ and $T(x)$ the corresponding return period[22]. Thus, an impact level with a cumulative probability of being exceeded by 10% of all modelled years refers to the impact of a 10-year return period ($T(x) = 10$ y =1/10% y$^{-1}$). In our example of the CanESM2, this corresponds to the year with the 100[th] largest impact within a given climate period ($v(x) = 10\%y^{-1} = 100*0.1\%y^{-1}$). (5) The exceedance frequencies shown in this study (Fig. 1), are expressed as median values over all SMILEs. Hence, each large ensemble is weighted equally, irrespective of its member size. Quantification of model agreement and uncertainties can be found in the supplementary information.

## Data availability

The SMILE climate model output is available via https://www.cesm.ucar.edu/projects/community-projects/MMLEA. The output of the 84-member ensemble of the CESM1.2 used in this analysis is available at https://data.iac.ethz.ch/Fischer_et_al_2021_RecordExtremes. A subset of daily mortality data is available at https://doi.org/10.48350/155666.

## Code availability

All code necessary to reproduce the analysis is made available on https://github.com/samluethi/ProbaHeat and permanently stored at https://doi.org/10.5281/zenodo.8074922.

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

## Acknowledgements

We acknowledge the US CLIVAR Working Group on Large Ensembles for supporting the Multi-Model Large Ensemble Archive. We thank the climate modelling groups for producing and making available their model output. We also acknowledge all members of the Multi-Country Multi-City (MCC) Collaborative Research Network for granting access to their data. E.F. acknowledges funding from the EU Horizon 2020 Project XAIDA (grant agreement 101003469) and by the Swiss National Science Foundation (grant 200020_178778). V.H. was supported by the European Union's Horizon 2020 research and innovation programme (Marie Skłodowska-Curie Grant Agreement No.: 101032087). J.K. and A.U. acknowledge funding from the Czech Science Foundation (project 22-24920 S).

## Author contributions

S.L., A.M.V.C., E.M.F, and D.N.B. were involved in conceptualization. S.L., A.M.V.C. and E.M.F. designed the methodology. S.L. conducted the formal analysis. A.M.V.C., E.M.F., B.A., M.D.S.Z.S.C., Y.L.G., Y.G., Y.H., V.H., J.K., E.L., D.R., N.R., N.S., S.S., A.U. and A.G. were involved in resources and data curation. S.L. undertook visualization. S.L., A.M.V.C., C.F. and N.S. wrote the draft manuscript. S.L., C.F., A.M.V.C., E.M.F., B.A., M.D.S.Z.S.C., Y.L.G., Y.G., Y.H., V.H., J.K., E.L., D.R., N.R., N.S., S.S., A.U., A.G., D.N.B. reviewed the manuscript. D.N.B. acquired funding. A.M.V.C. and D.N.B. supervised the project.

## Competing interests

The authors declare no competing interests.

## Additional information

[1]Institute for Environmental Decisions, ETH Zurich, Zurich, Switzerland. [2]Federal Office of Meteorology and Climatology MeteoSwiss, Zurich, Switzerland. [3]Institute for Atmospheric and Climate Science, ETH Zurich, Zurich, Switzerland. [4]Gangarosa Department of Environmental Health. Rollins School of Public Health, Emory University, Atlanta, GA, USA. [5]Department of Public Health Environments and Society, London School of Hygiene & Tropical Medicine, London, UK. [6]Department of Pathology, Faculty of Medicine, University of São Paulo, São Paulo, Brazil. [7]Environmental and Occupational Medicine, National Taiwan University (NTU) College of Medicine and NTU Hospital, Taipei, Taiwan. [8]National Institute of Environmental Health Science, National Health Research Institutes, Zhunan, Taiwan. [9]Graduate Institute of Environmental and Occupational Health Sciences, NTU College of Public Health, Taipei, Taiwan. [10]Climate, Air Quality Research Unit, School of Public Health and Preventive Medicine, Monash University, Melbourne, Australia. [11]Center for Climate Change Adaptation, National Institute for Environmental Studies, Tsukuba, Japan. [12]IBE-Chair of Epidemiology, LMU Munich, Munich, Germany. [13]Department of Physical, Chemical and Natural Systems, Universidad Pablo de Olavide, Sevilla, Spain. [14]Institute of Atmospheric Physics, Czech Academy of Sciences, Prague, Czech Republic. [15]Faculty of Environmental Sciences, Czech University of Life Sciences, Prague, Czech Republic. [16]School of Epidemiology & Public Health, Faculty of Medicine, University of Ottawa, Ottawa, ON, Canada. [17]Environmental Health Science and Research Bureau, Health Canada, Ottawa, ON, Canada. [18]CIBER of Epidemiology and Public Health, Madrid, Spain. [19]Center for Environmental and Respiratory Health Research (CERH), University of Oulu, Oulu, Finland. [20]Department of Epidemiology, Instituto Nacional de Saúde Dr. Ricardo Jorge, Lisbon, Portugal. [21]Centre for Statistical Methodology, London School of Hygiene & Tropical Medicine, London, UK. [22]Centre on Climate Change & Planetary Health, London School of Hygiene & Tropical Medicine, London, UK. [23]Institute of Social and Preventive Medicine, University of Bern, Bern, Switzerland. [24]Oeschger Center for Climate Change Research, University of Bern, Bern, Switzerland. ✉e-mail: samuel.luethi@usys.ethz.ch; anamaria.vicedo@ispm.unibe.ch

