## [Peer Review File · Nature Communications]

Rapid increase in the risk of heat-related mortalityREVIEWER COMMENTS

Reviewer #1 (Remarks to the Author):

Review of the manuscript "Rapid increase in the risk of heat-related mortality"

This study presented the future increase in heat-related mortality, so the significant reduction in return period using the large ensembles of several global model simulations. While I acknowledge that a timely topic can be of interest to a wide readership, I do wonder whether the material is novel enough to be published in Nature Communications. It was hard for me to see new findings in terms of both the conclusion and methodology. This manuscript does not appeal to me and I am not inclined to recommend it to be published in Nature Communications. More detailed comments are as follows.

1. If the heat-related mortality is the function of daily mean temperature solely, Fig.1 (g, h, i) seems to be a bit suspicious to me. I speculate that the large ensemble members perturbed by the initial conditions could not create such a large uncertainty range.
2. The empirical curves to represent the relationship between heat and mortality risk was established based on the daily mean temperature and mortality data. The statistical method called "quasi-Poisson regression time series analyses with distributed lag nonlinear models" that was developed by one of the authors was applied. While I admit that I am not an expert in the area of statistics or epidemiology, I do not think that daily mean temperature can be the best meteorological variable to measure the excess death due to heat. The maximum temperature may be a better indicator. Compared to mortality data, it is relatively easy to get more meteorological variables.
3. RCP8.5 scenario provides the transient emission forcing. But, 1.5°C and 2.0°C warming should represent the equilibrium state of that warming. There is no explanation.
4. I did not check the horizontal resolutions for individual GCMs, but most of GCMs have resolutions over 100 km. I was wondering how the coarse grid of GCMs can be matched with each city-level location over some regions (e.g., Western Europe, Japan) with dense locations. I think that it is possible for the same GCM grid to be selected for the different cities.

Reviewer #2 (Remarks to the Author):

This study assessed the future heat-related mortality burden in 748 locations from 47 countries. It is an interesting topic and might have important implications for the development of protective actions from climate change. However, I have several major comments.

1. Different from previous studies, the present paper assessed the future heat-related excess mortality burden using the probabilistic approach. The description of deterministic scenarios and the probabilistic approach is unclear in the current introduction, please elaborate these important points.
2. Results: In the first part of this section, only the results of São Paulo, Paris and Bangkok were presented. Please provide the reason for why these cities were selected.
3. The authors put the Result and Discussion section together. I would suggest a separate Discussion section, and the comparison with results from previous studies are needed to be strengthened. And the implications for the future research and the development of public health policies are also required to be improved.
4. Methods:
 - (1) Line 181, "In order to obtain warming-based reference periods of 0.7, 1.2, 1.5 and 2.0°C", it is better to provide the information on detailed period for each climate model reaching these warming targets.
 - (2) Lines 250-261, I find this part is difficult to follow. I would suggest take a typical example (like Paris) to make it more readable.
 - (3) Sensitivity analyses are required to test the robustness of the main results, such as the

placements of knots for temperature-mortality association, and degrees of freedom for time variable.

Reviewer #3 (Remarks to the Author):

This interdisciplinary work combines state-of-the-art tools from climate modelling with health data to quantify the impact of future changes in heat extremes on human health. The work is novel and highly relevant.

Overall, the manuscript is in a very good shape and I recommend publication following minor revisions that would help to clarify the text.

On a general note, I was wondering why you limit your analysis to five large ensembles. By now, more than ten SMILEs are available through the MMLEA and the ESGF archive. There are also several SMILEs using CMIP6 forcing, which you don't mention. In this context, I would appreciate a justification or discussion for the following points:

- Do all of the chosen models represent internal variability well? Did you evaluate this against observations? Could including models that don't represent observed variability well result in a too wide spread in your analysis? Why do you exclude the MPI-ESM-LE model from the MMLEA despite its large ensemble size?
- Are the selected models sufficient to capture response uncertainties? Is there a chance you underestimate the spread by not including more models?
- Figure 7: is the rapid increase in heat mortality for Paris in CESM1.2-LE an artefact of the analysis or model? Or is it a result that we should trust?

I 60: You argue that using cities as the unit of analysis makes the study more relevant for adaptation planning. But does the coarse resolution of the models justify this approach? I would imagine that the climate models don't represent e.g. heat island effects well, which would be important in bigger cities.

II. 104-106: 'albeit with uncertainty due to different warming rates of the climate models' Can you elaborate? Aren't warming levels used to make the analysis independent of the timing of reaching e.g. 1.5°C or 2°C? Or do you mean that a model with a larger warming rate would have a larger trend due to the forced response within a 20-year period?

I. 137: 'becoming normal' What do you define as normal here? Can you make this statement more quantitative?

II. 224–231 and 232–242 seem to be different revisions of the same paragraph. Can you please remove one of them?

Table S1: The formatting of city names should be cleaned. Most importantly, names seem to be randomly abbreviated which makes searching for a specific city difficult. (e.g. 'nasv is probably Nashville? 'philly'?) Also adding white space to names ('NewYork') and proper capitalization would improve readability.

Authors response to the invited major revisions

Research article: Rapid increase in the risk of heat-related mortality

Authors: Samuel Lüthi, Christopher Fairless, Erich M. Fischer, Noah Scovronick, Ben Armstrong, Micheline S. Z. S. Coelho, Yue Leon Guo, Yuming Guo, Yasushi Honda, Veronika Huber, Jan Kysely, Eric Lavigne, Dominic Royé, Niilo Rytö, Susana Silva, Alès Urban, Antonio Gasparrini, David N. Bresch, and Ana M. Vicedo-Cabrera

We thank the three anonymous referees for their highly constructive comments, which improve the quality of this manuscript. The original comments from the referees are listed below directly followed by our responses in blue incl. adjustments to the manuscript in **bold green**.

REVIEWER COMMENTS

Reviewer #1 (Remarks to the Author):

Review of the manuscript “Rapid increase in the risk of heat-related mortality”

This study presented the future increase in heat-related mortality, so the significant reduction in return period using the large ensembles of several global model simulations. While I acknowledge that a timely topic can be of interest to a wide readership, I do wonder whether the material is novel enough to be published in Nature Communications. It was hard for me to see new findings in terms of both the conclusion and methodology. This manuscript does not appeal to me and I am not inclined to recommend it to be published in Nature Communications. More detailed comments are as follows.

We thank the reviewer for the thorough review. We want to highlight the novelty of this interdisciplinary study, as we combine state-of-the art tools from epidemiology as well as from climate modelling. To our knowledge, no other study used a set of large climate model ensembles to quantify heat-mortality impacts of extreme seasons, including climate change projections. Furthermore, thanks to the MCC data consortium, we could rely on the (to our knowledge) largest mortality database for temperature-mortality studies.

1. If the heat-related mortality is the function of daily mean temperature solely, Fig.1 (g, h, i) seems to be a bit suspicious to me. I speculate that the large ensemble members perturbed by the initial conditions could not create such a large uncertainty range.

We thank the reviewer for this comment. The impact estimates account for the uncertainties both from the epidemiological analysis (depicted with the 95% CI Fig 1 a,b,c) and the model uncertainty in Fig. 1 (g, h, i) for the respective 1-in-100-year season. The epidemiological estimates (risks) induces a large uncertainty, especially when extrapolating the exposure-response curve to the unobserved temperature range.

However, we displayed the changes in the forced response, i.e. the model response in the absence of internal variability and did not illustrate the uncertainty induced by internal climate variability in this plot. To address this uncertainty, we estimate the uncertainty of a 1-in-100 year season using bootstrapping for each large ensemble. As the referee hypothesizes correctly, the uncertainty stemming from this variability generally induces less

uncertainty as compared to the empirical epidemiological uncertainty. We therefore adjust Fig. 1 (g, h, j) to reflect on all three sources of uncertainty (epidemiological, choice of model, and internal climate variability) by incorporating error bars with two whiskers: the inner to denote the uncertainty stemming from the climatic variability, the outer to denote the epidemiological uncertainty.

Figure 1: Update to the original Figure 1 g-i. New Figure includes two whiskers within the uncertainty bar to reflect on uncertainty from climatic variability (inner) and the epidemiological uncertainty (outer).

In addition to the adjustment of the Figure, we change the caption to:

Fig.1) [...] (g–i) Modelled magnitude of the annual heat mortality fraction from a 1-in-100 year season for different warming levels. Results are displayed for each SMILE (bars) including uncertainty estimates depicted with the 95% empirical confidence intervals **accounting for the internal climate variability (inner whisker) and imprecision of the exposure-response associations (outer whisker)**. [...]

And reflect this source of uncertainty also in the text at line 107-111:

This fraction would more than double in all three example cities if global mean temperatures rose from 0.7°C to 2.0°C above pre-industrial levels, albeit with uncertainty due to different warming rates of the climate models, **internal climate variability (inner whisker of the grey ticks) and the epidemiological uncertainty within the relative risk associations (outer whisker)**, as displayed by the shaded area (95% eCI) in Figure 1a-c. The internal climate variability is assessed by bootstrapping the ensemble members for each model-specific exceedance frequency curve.

2. The empirical curves to represent the relationship between heat and mortality risk was established based on the daily mean temperature and mortality data. The statistical method called “quasi-Poisson regression time series analyses with distributed lag nonlinear models” that was developed by one of the authors was applied. While I admit that I am not an expert in the area of statistics or epidemiology, I do not think that daily mean temperature can be the best meteorological variable to measure the excess death due to heat. The maximum temperature may be a better indicator. Compared to mortality data, it is relatively easy to get more meteorological variables.

There has been a relatively lively debate around the optimal variable to estimate heat mortality. For example, in a systematic evaluation of different variables, T_{mean} was found to be better suited to predict mortality compared to T_{max} (Xu et al., 2018). As the reviewer suggests, daily maximum temperature could be used, as well as daily minimum temperature (which would be an indicator for bodies reduced abilities to rest and recover during the night)

or other variables. This has been assessed in previous work using the same modelling approach (Armstrong et al., 2019; Gasparrini et al., 2015; Guo Yuming et al., 2017).

For the concrete case of T_{mean} vs. T_{max} , the small difference is due to the very strong linear relation of the two variables. Hence, exchanging one for the other yields a shift of the relative risk curve along the x-axis, as displayed in Figure 2 (Figure adopted from Ragettli et al., 2023) but does not affect the explained variance.

Figure : Relative risk curves for different temperature variables for Switzerland (here expressed as cumulative odds-ratio, which is the same as relative risk). Figure adopted from Ragettli et al. (2023).

To reflect on our choice of variable we suggest adding the following in the Method section at Lines 219-222 (also in combination with the point raised by reviewer #2):

Model specification and parameterisation is based on previous studies ^{17, 50, 51}, **as well as the choice of mean temperature as explanatory variable (Armstrong et al., 2019; Gasparrini et al., 2015; Guo Yuming et al., 2017; Xu et al., 2018) and tested for their sensitivity (Guo et al. (2017), Madaniyazi et al., (2022), Wu et al. (2022) and Huber et al. (2022)).**

3. RCP8.5 scenario provides the transient emission forcing. But, 1.5°C and 2.0°C warming should represent the equilibrium state of that warming. There is no explanation.

In line with the recent IPCC AR6 reports (IPCC, 2021), we communicate (future) climate impacts based on global warming levels. These are defined as the *first-time a given warming level is crossed for a 20-year period* in a transient simulation. The response of temperature-related variable at a given global warming level has been demonstrated to be very consistent across emission scenario (IPCC, 2021 (Chapter 11, in IPCC AR6 WG1). Thus, the reviewer is right that the warming levels are not meant to represent an equilibrium state. Thereby, it is consistent with the approach used in IPCC AR6 WG1, WG2 and Synthesis reports.

4. I did not check the horizontal resolutions for individual GCMs, but most of GCMs have resolutions over 100 km. I was wondering how the coarse grid of GCMs can be matched with each city-level location over some regions (e.g., Western Europe, Japan) with dense locations. I think that it is possible for the same GCM grid to be selected for the different cities.

Yes, the grid spacing of all GCMs used in this study is around 100 km or more. However, as described in the Methods, we apply bias correction against a local temperature time series to appropriately display the climatology of individual locations. Hence, even if two cities have the same nearest neighboring GCM grid-point, our approach will still be able to differentiate

the two (although climatic changes will be very similar). The temperature response to increased greenhouse gases on the other hand is relatively homogeneous in space and expected to be captured well even at GCM resolutions.

Reviewer #2 (Remarks to the Author):

This study assessed the future heat-related mortality burden in 748 locations from 47 countries. It is an interesting topic and might have important implications for the development of protective actions from climate change. However, I have several major comments.

We thank reviewer #2 for the suggestions and comments that clearly improved the understanding of our manuscript.

1. Different from previous studies, the present paper assessed the future heat-related excess mortality burden using the probabilistic approach. The description of deterministic scenarios and the probabilistic approach is unclear in the current introduction, please elaborate these important points.

Agreed, as the probabilistic component is really a core part of this study, it makes a lot of sense to introduce it with more clarity. We suggest incorporating the following sentences at Line 45:

While several studies have projected future heat-related mortality, they were based on a selected set of deterministic scenarios, **and estimated future heat mortality levels as the mean of a given climate period or scenario.**

And at line 50-52:

Such probabilistic risk assessments are widespread in the risk assessment of natural hazards, such as flooding (Arnell & Gosling, 2016) or tropical cyclones (Meiler et al., 2022), as knowledge of the magnitude and probability of potential impacts are key to prepare for and adapt to climatic extremes (e.g. Bresch & Aznar-Siguan, 2020).

2. Results: In the first part of this section, only the results of São Paulo, Paris and Bangkok were presented. Please provide the reason for why these cities were selected.

We aimed to highlight illustrative locations from different climatic zones and continents representing the variability (climate, population characteristics) across locations included in the analysis. We thus selected a location in a sub-tropical (Sao Paolo), tropical (Bangkok) and temperate (Paris) climate zone. We decided to restrict the selection to large cities with enough statistical power to derive stable exposure-response curves. Paris additionally yields the advantage that it is a well-studied city in terms of heat-mortality, due to the prominent 2003 heatwave.

To make this selection more transparent, we change the sentence at Line 92/93 to:

The impact exceedance frequency curves at each of the four different warming levels are displayed for three selected locations, representing different continents and climate zones, in Figure 1d-f.

3. The authors put the Result and Discussion section together. I would suggest a separate Discussion section, and the comparison with results from previous studies are needed to be

strengthened. And the implications for the future research and the development of public health policies are also required to be improved.

Taking into account the reviewer's suggestion, we now split our last section (Worsening the odds of heat-mortality) to add a separate Discussion. To embed our results in the literature we added the following two paragraphs (Lines 152-161):

In this work we combined state-of-the-art techniques from climate change epidemiology with the latest approaches in climate science to quantify extreme seasons. **Our results align well with related work although a direct comparison of numbers is hampered by different selection of reference scenarios, time scales or geographical scope (Lo et al., 2019; Mitchell et al., 2016). As an example, the change in return period of the mortality counts during the 2003 heatwave in Paris was estimated to decrease from 1-in-300 years, in a world with no anthropogenic climate change, to a 1-in-70 years in the actual climate of 2003 (Mitchell et al., 2016), which aligns well with our findings (Fig 1).**

Looking solely at climatic variables, the rapid reduction of the return period of such heatwaves were reported early on (Stott et al., 2004), estimating the 2003 temperature anomaly over Southern Europe to occur every other year (1-in-2 years return period) by 2040. Similarly, Christidis et al. (2015) found a tenfold decrease of the return period of extremely hot European summers between the 1990s and the early 2000s (from 1-in-50 years to 1-in-5 years), again aligning well with our findings. Similar results were also obtained using SMILE data (Suarez-Gutierrez et al., 2018), where the return period of a monthly temperature extreme of a 1-in-100 year event in a 1.5°C world is reduced to a roughly 1-in-10 year event in a 2°C world, whilst the new 1-in-100 year events of the 2°C world represent uncharted territories.

4. Methods:

(1) Line 181, "In order to obtain warming-based reference periods of 0.7, 1.2, 1.5 and 2.0°C", it is better to provide the information on detailed period for each climate model reaching these warming targets.

We suggest to include the following table in order to introduce more clarity:

Climate model	Number of members	Model resolution	Forcing	Model time period			
				2000 (0.7°C)	2020 (1.2°C)	1.5°C	2.0°C
CESM1.2	84	1.9°x2.5°	Hist / rcp8.5	1989-2008	2006-2025	2014-2033	2027-2046
CESM1-CAM5	40	1.3°x0.9°	Hist / rcp8.5	1990-2009	2007-2026	2015-2034	2028-2047
CanESM2	50	2.8°x2.8°	Hist / rcp8.5	1982-2001	1995-2014	2003-2022	2016-2035
GFDL-ESM2M	30	2.0°x2.5°	Hist / rcp8.5	1988-2007	2010-2029	2024-2043	2042-2061
CSIRO-Mk3.6.0	30	1.9°x1.9°	Hist / rcp8.5	1996-2015	2014-2033	2022-2041	2035-2054

We adjust the paragraph describing the climate model data accordingly (lines 194 ff.) and removed the ensemble member size.

(2) Lines 250-261, I find this part is difficult to follow. I would suggest take a typical example (like Paris) to make it more readable.

Thanks a lot for this suggestion. We agree that this paragraph was not easy to follow. In the revised manuscript we use a SMILE as example rather than a location to make it more readable. We thus reformulate the paragraph to (Lines 260 ff.):

Return levels and return periods of heat-related mortality in each location and for each SMILE are empirically estimated in the following way: (1) Daily heat-related mortality numbers are calculated for each ensemble member using the approach described above and aggregated to annual levels (2) This yields $N = n_{\text{member}} \times 20$ annual heat-related mortality impacts for each 20-yr period. **E.g. for the CanESM2 ensemble this results in a sample size of $N = 1,000 (= 50 \times 20)$ possible years.** (3) The empirical probability of occurrence for any given year in this period can thus be expressed as $p = 1/N$, as all of these years can be treated with equal probability of occurrence and are independent of one-another. **Hence, within the CanESM2, each modelled year has a probability of occurrence of 0.1% (= 1 / 1000)** (4) The return periods are then quantified by calculating the cumulative probabilities of exceedance of impact levels: $v(x)=1/T(x)$, where $v(x)$ is the exceedance frequency of impact x and $T(x)$ the corresponding return period. Thus, an impact level with a cumulative probability of being exceeded by 10% of all modelled years refers to the impact of a 10-year return period. **In our example of the CanESM2, this corresponds to the year with the 100th largest impact within a given climate period** (5) The exceedance frequencies shown in this study (Figure 1), are expressed as median values over all SMILEs. Hence, each large ensemble is weighted equally, irrespective of its member size. Quantification of model agreement and uncertainties can be found in the extended data

(3) Sensitivity analyses are required to test the robustness of the main results, such as the placements of knots for temperature-mortality association, and degrees of freedom for time variable.

The parameterization of the epidemiological model has been continuously tested and improved over the past couple of years (i.e.(Gasparrini et al., 2010, 2015, 2017)) and became the standard model choice for several studies over the past years (i.e. (de Schrijver et al., 2021; Huber et al., 2022; Scovronick et al., 2018; Vicedo-Cabrera et al., 2018). Furthermore, an extensive sensitivity analysis was performed i.e. by Guo et al. (2017), Madaniyazi et al., (2022), Wu et al. (2022) and Huber et al. (2022) and found that this set of parameterization is indeed optimal (Huber et al., 2022) and that differences are not substantial (Wu et al., 2022).

Still, as suggested by the reviewer, we changed the parameterization for the location of the knots and the degrees of freedom per season, as displayed in Fig. 2 and Fig. 3 underneath. The newly obtained exposure-response curves (displayed in red) fall well into the empirical confidence interval that we obtained from the Monte-Carlo sampling of the spline parameters (with the only slight exception of Paris and knots at (10,80,95)).

We thus suggest to refer – in combination with the point raised by reviewer #1 - to related work (Line 219-222):

Model specification and parameterisation is based on previous studies ^{17, 50, 51}, **as well as the choice of mean temperature as explanatory variable (Armstrong et al., 2019; Gasparrini et al., 2015; Guo Yuming et al., 2017) and tested for their sensitivity (Guo et al. (2017), Madaniyazi et al., (2022), Wu et al. (2022) and Huber et al. (2022)).**

Figure 2: Sensitivity analysis of placement of internal knots for the relative risk curves of São Paulo (left column), Paris (middle) and Bangkok (right), corresponding to Figure 1a-c. Original knots used in the analysis are placed at the 10th, 75th and 90th percentile.

Figure 3: Sensitivity analysis of selection of degrees of freedom from seasonality for the relative risk curves of São Paulo (left column), Paris (middle) and Bangkok (right), corresponding to Figure 1a-c. Original degrees of freedom used in the analysis is 8.

Reviewer #3 (Remarks to the Author):

This interdisciplinary work combines state-of-the-art tools from climate modelling with health data to quantify the impact of future changes in heat extremes on human health. The work is novel and highly relevant.

Overall, the manuscript is in a very good shape and I recommend publication following minor revisions that would help to clarify the text.

We would also like to express our gratitude to referee #3 for the thorough review.

On a general note, I was wondering why you limit your analysis to five large ensembles. By now, more than ten SMILEs are available through the MMLEA and the ESGF archive. There are also several SMILEs using CMIP6 forcing, which you don't mention. In this context, I would appreciate a justification or discussion for the following points:

- Do all of the chosen models represent internal variability well? Did you evaluate this against observations? Could including models that don't represent observed variability well result in a too wide spread in your analysis? Why do you exclude the MPI-ESM-LE model from the MMLEA despite its large ensemble size?

As the focus of this study lies on heat mortality, we did not systematically evaluate the raw climate model output of the different SMILEs for climatic variability. However, all the five selected large ensembles capture the observational heat mortality exceedance frequency curves well within their 95% confidence interval (see Fig. 6 in the manuscript).

We bias correct the climate model output using quantile-mapping. Hence, we adjust for biases in mean as well in variability (Fig 4). Thus, we expect that the internal variability is adequately represented.

Figure 4: Temperature distribution for Paris. Observational time series (2000-2015, black line), with raw model output of the nearest-neighbor grid point for all 84 ensemble members of CESM1.2-LE (individual blue lines, left-hand side); after bias correction (turquoise lines, middle); and after bias correction at global warming levels of 1.5°C (violet lines, right-hand side).

- Are the selected models sufficient to capture response uncertainties? Is there a chance you underestimate the spread by not including more models?

From looking at Lehner et al., 2020 (Figure S1, displayed below for convenience), our model selection includes models with high variability and others with relatively low variability and is not systemically biased in terms of global temperature variability. Furthermore, the variability

(standard deviation) of the mentioned MPI-ESM-LE falls well within the range of our model selection (CESM1-CAM5, CanESM2, CSIRO-Mk3-6-0, GFDL-ESM2M).

Figure 5: Figure adapted from (Lehner et al., 2020)

Finally, as all displayed results focus on median impacts, we don't expect adding an additional climate model to substantially change any results. The MPI-ESM-LE was not included due to the lack of daily output in our local archive.

- Figure 7: is the rapid increase in heat mortality for Paris in CESM1.2-LE an artefact of the analysis or model? Or is it a result that we should trust?

The strong jump at the tail of the exceedance frequency curve from the CESM1.2 LE for Paris is indeed intriguing and could be interpreted as a *gray swan* event. Such jumps are also present within other LEs at other locations. We noticed this behavior as well during the write up of the study. However, we deliberately kept our focus on the 1-in-100 year events, which we are able to quantify in a more stable fashion, including associated probabilities.

For the mentioned extremes a storyline-type approach of quantification, including analysis of weather patterns, would be more suitable, which – in our view – would distort the focus of this study. But this could result in an interesting focus for a follow up study.

Still, for illustration – the mentioned extreme season is characterized by a nearly 60-day period with mean temperatures over 30°C, including 4 peaks. These peaks, have a strong impact on the mortality counts, due to the strong non-linearity in the Paris heat-mortality curve.

I 60: You argue that using cities as the unit of analysis makes the study more relevant for adaptation planning. But does the coarse resolution of the models justify this approach? I

would imagine that the climate models don't represent e.g. heat island effects well, which would be important in bigger cities.

It is true, that effects of urban heat islands are not explicitly represented in this study although the CESM2 model accounts for urban surface and represents the respective radiative and turbulent fluxes at the subgrid-scale level. However, the bias adjustment implicitly accounts for part of the urban heat island effects and in principle can account for the affect as long as the urban heat island effect results in a systematic offset.

Note that we are limited by the availability of mortality data, which is geographically aggregated to city-scales. Hence, even if we had climate data which can represent urban features, we could still not make full use of this data, as we don't know *where* in the city the fatalities occurred.

To reflect on this important remark in the manuscript, we suggest to add the following two sentences in the discussion about or limitations in thus study (Lines 169-170):

Also important small-scale climatic differences, such as urban heat island effects are not resolved (Oke, 1982). This is potentially problematic, as they can overlap with vulnerable communities (Hoffman et al., 2020).

II. 104-106: 'albeit with uncertainty due to different warming rates of the climate models' Can you elaborate? Aren't warming levels used to make the analysis independent of the timing of reaching e.g. 1.5°C or 2°C? Or do you mean that a model with a larger warming rate would have a larger trend due to the forced response within a 20-year period?

We identify two relevant factors, how different warming rates are still present: (1) The one you mention, that the difference within the first and the last year within a climate period is somewhat larger for models with a stronger rate of warming. And (2), that we apply SMILE specific bias-correction, which hence corrects LEs in a different fashion.

Due to the remarks of reviewer #1, we already adjusted this section to also reflect on climatic variability and epidemiological uncertainty.

I. 137: 'becoming normal' What do you define as normal here? Can you make this statement more quantitative?

We reformulate to:

We want to highlight two main findings from our study. First, we found that what used to be extreme (1-in-100 year) heat-mortality seasons are becoming **frequent, and need to be expected every 2 - 5 years in most locations.**

II. 224–231 and 232–242 seem to be different revisions of the same paragraph. Can you please remove one of them?

Thanks a lot for spotting – this must have been a mistake during the internal reviews.

We delete the lines 232-237.

Table S1: The formatting of city names should be cleaned. Most importantly, names seem to be randomly abbreviated which makes searching for a specific city difficult. (e.g. 'nasv is probably Nashville? 'philly'?) Also adding white space to names ('NewYork') and proper capitalization would improve readability.

We've cleaned and adjusted the table on quite many instances. Thanks a lot for raising this point. As there were many adjustment, we refrain from listing each one separately here.

References

- Armstrong, B., Sera, F., Vicedo-Cabrera, A. M., Abrutzky, R., Åström, D. O., Bell, M. L., Chen, B.-Y., de Sousa Zanotti Stagliorio Coelho, M., Correa, P. M., Dang, T. N., Diaz, M. H., Dung, D. V., Forsberg, B., Goodman, P., Guo, Y.-L. L., Guo, Y., Hashizume, M., Honda, Y., Indermitte, E., ... Gasparrini, A. (2019). The Role of Humidity in Associations of High Temperature with Mortality: A Multicountry, Multicity Study. *Environmental Health Perspectives*, 127(9), 097007. <https://doi.org/10.1289/EHP5430>
- Arnell, N. W., & Gosling, S. N. (2016). The impacts of climate change on river flood risk at the global scale. *Climatic Change*, 134(3), 387–401. <https://doi.org/10.1007/s10584-014-1084-5>
- Bresch, D. N., & Aznar-Siguan, G. (2020). CLIMADA v1.4.1: Towards a globally consistent adaptation options appraisal tool. *Geoscientific Model Development Discussions*, 1–20. <https://doi.org/10.5194/gmd-2020-151>
- Christidis, N., Jones, G. S., & Stott, P. A. (2015). Dramatically increasing chance of extremely hot summers since the 2003 European heatwave. *Nature Climate Change*, 5(1), Article 1. <https://doi.org/10.1038/nclimate2468>
- de Schrijver, E., Folly, C. L., Schneider, R., Royé, D., Franco, O. H., Gasparrini, A., & Vicedo-Cabrera, A. M. (2021). A Comparative Analysis of the Temperature-Mortality Risks Using Different Weather Datasets Across Heterogeneous Regions. *GeoHealth*, 5(5), e2020GH000363. <https://doi.org/10.1029/2020GH000363>
- Gasparrini, A., Armstrong, B., & Kenward, M. G. (2010). Distributed lag non-linear models. *Statistics in Medicine*, 29(21), 2224–2234. <https://doi.org/10.1002/sim.3940>
- Gasparrini, A., Guo, Y., Hashizume, M., Lavigne, E., Zanobetti, A., Schwartz, J., Tobias, A., Tong, S., Rocklöv, J., Forsberg, B., Leone, M., De Sario, M., Bell, M. L., Guo, Y.-L. L., Wu, C., Kan, H., Yi, S.-M., de Sousa Zanotti Stagliorio Coelho, M., Saldiva, P. H. N., ... Armstrong, B. (2015). Mortality risk attributable to high and low ambient temperature: A multicountry observational study. *The Lancet*, 386(9991), 369–375. [https://doi.org/10.1016/S0140-6736\(14\)62114-0](https://doi.org/10.1016/S0140-6736(14)62114-0)
- Gasparrini, A., Guo, Y., Sera, F., Vicedo-Cabrera, A. M., Huber, V., Tong, S., de Sousa Zanotti Stagliorio Coelho, M., Nascimento Saldiva, P. H., Lavigne, E., Matus Correa, P., Valdes Ortega, N., Kan, H., Osorio, S., Kysely, J., Urban, A., Jaakkola, J. J. K., Rytö, N. R. I., Pascal, M., Goodman, P. G., ... Armstrong, B. (2017). Projections of temperature-related excess mortality under climate change scenarios. *The Lancet Planetary Health*, 1(9), e360–e367. [https://doi.org/10.1016/S2542-5196\(17\)30156-0](https://doi.org/10.1016/S2542-5196(17)30156-0)
- Guo Yuming, Gasparrini Antonio, Armstrong Ben G., Tawatsupa Benjawan, Tobias Aurelio, Lavigne Eric, Coelho Micheline de Sousa Zanotti Stagliorio, Pan Xiaochuan, Kim Ho, Hashizume Masahiro, Honda Yasushi, Guo Yue-Liang Leon, Wu Chang-Fu, Zanobetti Antonella, Schwartz Joel D., Bell Michelle L., Scortichini Matteo, Michelozzi Paola, Punnasiri Kornwipa, ... Tong Shilu. (2017). Heat Wave and Mortality: A Multicountry, Multicommunity Study. *Environmental Health Perspectives*, 125(8), 087006. <https://doi.org/10.1289/EHP1026>
- Hoffman, J. S., Shandas, V., & Pendleton, N. (2020). The Effects of Historical Housing Policies on Resident Exposure to Intra-Urban Heat: A Study of 108 US Urban Areas. *Climate*, 8(1), Article 1. <https://doi.org/10.3390/cli8010012>

- Huber, V., Ortiz, C. P., Puyol, D. G., Lange, S., & Sera, F. (2022). Evidence of rapid adaptation integrated into projections of temperature-related excess mortality. *Environmental Research Letters*, 17(4), 044075. <https://doi.org/10.1088/1748-9326/ac5dee>
- IPCC. (2021). *Summary for Policymakers. In: Climate Change 2021: The Physical Science Basis. Contribution of Working Group I to the Sixth Assessment Report of the Intergovernmental Panel on Climate Change [Masson-Delmotte, V., P. Zhai, A. Pirani, S.L. Connors, C. Péan, S. Berger, N. Caud, Y. Chen, L. Goldfarb, M.I. Gomis, M. Huang, K. Leitzell, E. Lonnoy, J.B.R. Matthews, T.K. Maycock, T. Waterfield, O. Yelekçi, R. Yu, and B. Zhou (eds.)].* 3–32. <https://doi.org/10.1017/9781009157896.001>.
- Lehner, F., Deser, C., Maher, N., Marotzke, J., Fischer, E. M., Brunner, L., Knutti, R., & Hawkins, E. (2020). Partitioning climate projection uncertainty with multiple large ensembles and CMIP5/6. *Earth System Dynamics*, 11(2), 491–508. <https://doi.org/10.5194/esd-11-491-2020>
- Lo, Y. T. E., Mitchell, D. M., Gasparrini, A., Vicedo-Cabrera, A. M., Ebi, K. L., Frumhoff, P. C., Millar, R. J., Roberts, W., Sera, F., Sparrow, S., Uhe, P., & Williams, G. (2019). Increasing mitigation ambition to meet the Paris Agreement's temperature goal avoids substantial heat-related mortality in U.S. cities. *Science Advances*, 5(6), eaau4373. <https://doi.org/10.1126/sciadv.aau4373>
- Madaniyazi, L., Armstrong, B., Chung, Y., Ng, C. F. S., Seposo, X., Kim, Y., Tobias, A., Guo, Y., Sera, F., Honda, Y., Gasparrini, A., Hashizume, M., & Multi-Country Multi-City (MCC) Collaborative Research Network. (2022). Seasonal variation in mortality and the role of temperature: A multi-country multi-city study. *International Journal of Epidemiology*, 51(1), 122–133. <https://doi.org/10.1093/ije/dyab143>
- Meiler, S., Vogt, T., Bloemendaal, N., Ciullo, A., Lee, C.-Y., Camargo, S. J., Emanuel, K., & Bresch, D. N. (2022). Intercomparison of regional loss estimates from global synthetic tropical cyclone models. *Nature Communications*, 13(1), Article 1. <https://doi.org/10.1038/s41467-022-33918-1>
- Mitchell, D., Heaviside, C., Vardoulakis, S., Huntingford, C., Masato, G., Guillod, B. P., Frumhoff, P., Bowery, A., Wallom, D., & Allen, M. (2016). Attributing human mortality during extreme heat waves to anthropogenic climate change. *Environmental Research Letters*, 11(7), 074006. <https://doi.org/10.1088/1748-9326/11/7/074006>
- Oke, T. R. (1982). The energetic basis of the urban heat island. *Quarterly Journal of the Royal Meteorological Society*, 108(455), 1–24. <https://doi.org/10.1002/qj.49710845502>
- Ragettli, M. S., Saucy, A., Flückiger, B., Vienneau, D., de Hoogh, K., Vicedo-Cabrera, A. M., Schindler, C., & Rössli, M. (2023). Explorative Assessment of the Temperature–Mortality Association to Support Health-Based Heat-Warning Thresholds: A National Case-Crossover Study in Switzerland. *International Journal of Environmental Research and Public Health*, 20(6), Article 6. <https://doi.org/10.3390/ijerph20064958>
- Scovronick, N., Sera, F., Acquaotta, F., Garzena, D., Fratianni, S., Wright, C. Y., & Gasparrini, A. (2018). The association between ambient temperature and mortality in South Africa: A time-series analysis. *Environmental Research*, 161, 229–235. <https://doi.org/10.1016/j.envres.2017.11.001>
- Stott, P. A., Stone, D. A., & Allen, M. R. (2004). Human contribution to the European heatwave of 2003. *Nature*, 432(7017), Article 7017. <https://doi.org/10.1038/nature03089>

- Suarez-Gutierrez, L., Li, C., Müller, W. A., & Marotzke, J. (2018). Internal variability in European summer temperatures at 1.5 °C and 2 °C of global warming. *Environmental Research Letters*, *13*(6), 064026. <https://doi.org/10.1088/1748-9326/aaba58>
- Vicedo-Cabrera, A. M., Guo, Y., Sera, F., Huber, V., Schleussner, C.-F., Mitchell, D., Tong, S., Coelho, M. de S. Z. S., Saldiva, P. H. N., & Lavigne, E. (2018). Temperature-related mortality impacts under and beyond Paris Agreement climate change scenarios. *Climatic Change*, *150*(3–4), 391–402.
- Wu, Y., Li, S., Zhao, Q., Wen, B., Gasparrini, A., Tong, S., Overcenco, A., Urban, A., Schneider, A., Entezari, A., Vicedo-Cabrera, A. M., Zanobetti, A., Analitis, A., Zeka, A., Tobias, A., Nunes, B., Alahmad, B., Armstrong, B., Forsberg, B., ... Guo, Y. (2022). Global, regional, and national burden of mortality associated with short-term temperature variability from 2000–19: A three-stage modelling study. *The Lancet. Planetary Health*, *6*(5), e410–e421. [https://doi.org/10.1016/S2542-5196\(22\)00073-0](https://doi.org/10.1016/S2542-5196(22)00073-0)
- Xu, Z., Cheng, J., Hu, W., & Tong, S. (2018). Heatwave and health events: A systematic evaluation of different temperature indicators, heatwave intensities and durations. *Science of The Total Environment*, *630*, 679–689. <https://doi.org/10.1016/j.scitotenv.2018.02.268>

REVIEWERS' COMMENTS

Reviewer #1 (Remarks to the Author):

I acknowledge the authors' efforts to revise the manuscript and most of my concerns have been addressed.

However, I still have reservations regarding the response to the doubt of the "novelty" of this work.

I do not believe that combining two existing methodologies can create additional novelty.

The authors may want to emphasize the probabilistic assessment using a large ensemble.

But, I do not think that the "actual" advantage of the probabilistic assessment over deterministic estimation has been clearly described.

In addition, I could not find any new findings in the abstract that have not been reported in other studies.

In particular, the last sentence in the abstract is too generic and lacks specificity of this study.

I can see this kind of sentence everywhere.

Reviewer #2 (Remarks to the Author):

The authors have addressed most of my comments. But there are other points needed to be well considered.

1. In the "Impact exceedance frequency curves" of the Method section, I find it difficult for me to follow the 4th point. Please use the data from Can ESM2 to clearly articulate this issue.

2. In panel (d-f) of Figure 2, it is quite hard to distinguish the lines for the baseline period, Risk 2000 and Risk 2020.

3. In the Discussion section, please discuss the possible seasons on the spatial variations in the return periods across the globe. And the implications for future research and the development of public health policies are still needed to be improved.

Second authors response to the invited final revisions

Research article: Rapid increase in the risk of heat-related mortality

Authors: Samuel Lüthi, Christopher Fairless, Erich M. Fischer, Noah Scovronick, Ben Armstrong, Micheline S. Z. S. Coelho, Yue Leon Guo, Yuming Guo, Yasushi Honda, Veronika Huber, Jan Kysely, Eric Lavigne, Dominic Royé, Niilo Ryti, Susana Silva, Alès Urban, Antonio Gasparrini, David N. Bresch, and Ana M. Vicedo-Cabrera

We thank the two anonymous referees for the second round of comments, which improve the quality of this manuscript. The original comments from the referees are listed below directly followed by our responses in **blue** incl. adjustments to the manuscript in **bold green**.

REVIEWER COMMENTS

Reviewer #1 (Remarks to the Author):

I acknowledge the authors' efforts to revise the manuscript and most of my concerns have been addressed. However, I still have reservations regarding the response to the doubt of the "novelty" of this work. I do not believe that combining two existing methodologies can create additional novelty.

We thank Reviewer #1 for the second round of comments. In contrast to the referee, we do think that combining two existing approaches from different disciplines can yield substantial novelty and added value. In our example, the non-linear location specific vulnerabilities change (continental) patterns of return periods of extreme season, such as displayed by the inland-coastal gradient over the US or the North-South gradient over Europe (as highlighted in the Results chapter). Furthermore, we think that communicating possible impacts (fatalities) of extreme seasons is more actionable for decision makers than a pure climatological view.

The authors may want to emphasize the probabilistic assessment using a large ensemble. But I do not think that the "actual" advantage of the probabilistic assessment over deterministic estimation has been clearly described.

We elaborated on the advantages of a probabilistic approach already based on the comments raised by reviewer #2 in the first round of reviews. The advantages of a probabilistic risk analysis are described in the introduction on the lines 46-52 and 57-62.

In addition, I could not find any new findings in the abstract that have not been reported in other studies. In particular, the last sentence in the abstract is too generic and lacks specificity of this study. I can see this kind of sentence everywhere.

We want to highlight that we modelled changes in return period of extreme heat mortality seasons and not of temperature alone. To our knowledge, this has not been reported by other studies on such a broad geographical scope and for different warming levels.

Reviewer #2 (Remarks to the Author):

The authors have addressed most of my comments. But there are other points needed to be well considered.

We want to thank Reviewer #2 for the constructive feedback which improved our discussion.

1. In the “Impact exceedance frequency curves” of the Method section, I find it difficult for me to follow the 4th point. Please use the data from Can ESM2 to clearly articulate this issue.

As suggested by the reviewer we propagated our numerical example of CanESM2 through the 4th point (Line 279 ff.):

(4) The return periods are then quantified by calculating the cumulative probabilities of exceedance of impact levels: $v(x) = 1/T(x)$, where $v(x)$ is the exceedance frequency of impact x and $T(x)$ the corresponding return period. Thus, an impact level with a cumulative probability of being exceeded by 10% of all modelled years refers to the impact of a 10-year return period ($T(x) = 10y = 1/10\% y^{-1}$). In our example of the CanESM2, this corresponds to the year with the 100th largest impact within a given climate period ($v(x) = 10\%y^{-1} = 100 \cdot 0.1\%y^{-1}$).

2. In panel (d-f) of Figure 2, it is quite hard to distinguish the lines for the baseline period, Risk 2000 and Risk 2020.

We adjusted the 2000 line to a more intense blue so that the contrast becomes stronger.

3. In the Discussion section, please discuss the possible seasons on the spatial variations in the return periods across the globe. And the implications for future research and the development of public health policies are still needed to be improved.

Under the assumption that the referee meant possible *reasons* (and not seasons) of spatial variations we included the following (Lines 163 ff.):

Our analysis identifies several regions with a pronounced shortening of return periods of extreme seasons. For tropical regions this is largely due to the small seasonality and year-to-year variability of temperature which therefore leads to large shifts in return periods of extremes in a hotter climate (Fischer & Knutti, 2015). In Southern Europe, Japan and along the US Atlantic and Gulf coast, the projected reduction in return periods is furthermore driven by demographic influences, such as the aging of societies and the heightened vulnerability of populations to heat.

To highlight implications for future research, we highlight some of the limitations of this study more clearly and thus included the following sentence on Line 189:

These points mark relevant areas for future research, especially the incorporation of changes in population as well as adaptation into future projections of heat mortality levels.

Thanks for suggesting incorporating implications for public health policies – this makes the findings of this study more actionable. We included the following sentences (168 ff.):

These results highlight a need to incorporate possible extreme scenarios and storylines of unprecedented heatwaves into the planning of public health policies as the experience from mortality impacts of past summer seasons is likely to underestimate the actual risk of heat mortality in the rapidly changing climate. Currently, most (European) heat-health warning systems focus on issuing warnings to relevant authorities and vulnerable people during or slightly before the onset of a heatwave (Casanueva et al., 2019). However, in contrast to other climatic extremes, such as floods, only few authorities systematically plan for rare but extreme scenarios.

References

- Casanueva, A., Burgstall, A., Kotlarski, S., Messeri, A., Morabito, M., Flouris, A. D., Nybo, L., Spirig, C., & Schwierz, C. (2019). Overview of Existing Heat-Health Warning Systems in Europe. *International Journal of Environmental Research and Public Health*, 16(15), Article 15. <https://doi.org/10.3390/ijerph16152657>
- Fischer, E. M., & Knutti, R. (2015). Anthropogenic contribution to global occurrence of heavy-precipitation and high-temperature extremes. *Nature Climate Change*, 5(6), Article 6. <https://doi.org/10.1038/nclimate2617>